# Widefield and Ultra-Widefield Retinal Imaging: A Geometrical Analysis

**DOI:** 10.3390/life13010202

**Published:** 2023-01-10

**Authors:** Amedeo Lucente, Andrea Taloni, Vincenzo Scorcia, Giuseppe Giannaccare

**Affiliations:** 1Private Practice, Studio Lucente, 87012 Castrovillari, Italy; 2Department of Ophthalmology, University “Magna Græcia”, 87012 Catanzaro, Italy

**Keywords:** widefield, ultra-widefield, imaging, retina, field of view, diabetic retinopathy

## Abstract

Diabetic retinopathy (DR) often causes a wide range of lesions in the peripheral retina, which can be undetected when using a traditional fundus camera. Widefield (WF) and Ultra-Widefield (UWF) technologies aim to significantly expand the photographable retinal field. We conducted a geometrical analysis to assess the field of view (FOV) of WF and UWF imaging, comparing it to the angular extension of the retina. For this task, we shot WF images using the Zeiss Clarus 500 fundus camera (Carl Zeiss Meditec, Jena, Germany). Approximating the ocular bulb to an ideal sphere, the angular extension of the theoretically photographable retinal surface was 242 degrees. Performing one shot, centered on the macula, it was possible to photograph a retinal surface of ~570 mm^2^, with a FOV of 133 degrees. Performing four shots with automatic montage, we obtained a retinal surface area of ~1100 mm^2^ and an FOV of 200 degrees. Finally, performing six shots with semi-automatic montage, we obtained a retinal surface area of ~1400 mm^2^ and an FOV of 236.27 degrees, which is close to the entire surface of the retina. WF and UWF imaging allow the detailed visualization of the peripheral retina, with significant impact on the diagnosis and management of DR.

## 1. Introduction

The last decades of the past century marked the development and plateau of retinal photography, with the commercialization of numerous devices. Retinal imaging obtained by a traditional fundus camera or by a Scanning Laser Ophthalmoscope (SLO) typically has an angular field of view (FOV) of 30 to 50 degrees, allowing optimal visualization of the posterior pole, but it is not able to capture the peripheral retina beyond the retinal vascular branches in a single shot. Photographic montage can be performed to show a larger field of view, but manual photomontage is not easy to achieve, and often falls short of expectations [1]. On the one hand, the clinician must rely on patient cooperation and fixation stability. On the other hand, several artifacts can affect the final result: retinal brightness is usually uneven in different retinal sectors, and the continuity of vessels along the vascular arches is often lacking. Over time, devices have integrated the automatic or semi-automatic montage of central and peripheral images, improving the overall quality [2].

The peripheral retina can show a wide variety of lesions, both primary and secondary to specific retinal affections. The most urgent diagnostic need towards wide-field imaging has been expressed in the Early Treatment Diabetic Retinopathy Study (ETDRS) [3]. This was one of the first multicenter clinical studies to take advantage of what was later defined as seven-standard-field (7SF) imaging, which covers an FOV of 75 degrees through a montage of seven 30-degree shots. This method significantly improved diagnostic performance, allowing for earlier and more appropriate treatment of diabetic retinopathy (DR). Since then, 7SF has been considered the gold standard for retinal photography, not only in DR screening, but in many other chorioretinal diseases [4].

Recent advances in optics allowed significant expansion of the photographable retinal field using the Widefield (WF) and Ultra-Widefield (UWF) technologies. Initially, a retinal photograph was defined as WF if it covered an FOV ≥ 50 degrees an UWF if it covered an FOV ≥ 100 degrees. The concept of WF imaging was not exclusively applied to fundus cameras: Toslak et al. constructed a portable prototype of a trans-pars-planar illuminator for UWF pediatric fundus photography, obtaining an FOV of 200 degrees [5].

Despite the increasing number of publications about WF and UWF technologies, retinal imaging and FOV measurement still remain controversial topics in the literature, with ambiguous terminology and a lack of standardized methods of assessment.

In 2019, the International Widefield Imaging Study Group (IWFISG) clarified the definitions to be followed for fundus photography [6]: (1) the posterior pole was identified as the retinal area up to the outer margin of the vascular branches; (2) midperiphery was defined as the retinal area up to the posterior border of the vorticose veins; and (3) far periphery was defined as the retinal area anterior to the vorticose veins.

According to The Royal College of Ophthalmologists [7], images can be classified as WF or UWF if they are captured as single shots, centered on the fovea, without being assembled in a photomontage; WF images must capture the midperiphery, up to the vorticose veins; UWF images must include the far periphery, beyond the vorticose veins. These rules also apply to imaging modalities other than the fundus photograph, such as fundus autofluorescence (FA) and optical coherence tomography (OCT).

FOV assessment in WF retinal imaging is particularly challenging. First, there are different measurement methods because the FOV angle can be subtended at different points on the optical axis. Furthermore, input visual angles project different output angles onto the retina, depending on the inclination of light rays with respect to the optical axis [8]. When measuring retinal linear distances, a 1:1 angular scaling is only possible in cases with small input angles, close to the optical axis, while scaling becomes non-linear when considering wider angles. Finally, reproducing an approximately spherical structure such as the eye on a flat surface inevitably leads to measurement errors and inaccuracies, which are particularly impactful in the analysis of WF imaging [9]. The purpose of this paper is to conduct a geometrical analysis of the FOV of WF and UWF imaging and compare it to the angular extension of the theoretically photographable retina in order to understand the clinical value of the additional findings that these new technologies may provide to the ophthalmologist, with a particular reference to DR.

## 2. Materials and Methods

Several methods can be used to assess FOV in retinal imaging, according to the point of the optical axis at which the FOV angle is subtended [10]. In traditional fundus cameras, quantitative evaluation of the FOV is based on the visual-angle (θv), defined as the angle subtended by the imaged retinal region at the center of the exit pupil of the eye. The maximum FOV obtainable using this method is almost 180 degrees (Figure 1A). The technical committee ISO/TC172/SC7 (Ophthalmic optics and instruments) of the International Organization for Standardization selected this as the standard method in the ISO 10940:2009. Together with ISO 15004-1 and ISO 15004-2, it specifies requirements and test methods for a fundus camera. ISO standards are revised and updated every 5 years. ISO 10940 was confirmed in 2019 [11]. In the majority of WF devices, the FOV is calculated using the eye-angle (θe), defined as the angle subtended by the imaged retinal region at the spherical center of the eye, that is, the point of insertion between the vertical diameter of the eyeball and the visual axis [10]. With this method, the FOV angle is subtended at a point closer to the retina. Consequentially, even when considering the same photographed retinal area, the FOV is higher than corresponding ISO values. In fact, the FOV can be wider than 180 degrees, extending far beyond the bulbar equator (Figure 1B).

The existence of different methods of measurement may lead to preventable mistakes and misunderstandings. In our geometrical analysis, we always refer to the eye-angle because it is the most widely adopted, both in the scientific literature and technical specifications of WF devices. In order to compare the FOV of WF and UWF imaging to the angular extension of the retina, we used the Zeiss Clarus 500 fundus camera (Carl Zeiss Meditec, Jena, Germany). Linear and area measures were calculated using the updated version of the built-in software provided with the device. Comparison between the theoretically photographable retina and images acquired by the fundus camera was performed on 4 eyes of 4 different male patients (age range: 25–36 years) without any ocular disease, with a spherical equivalent defect between −0.25 and +0.25 diopters. Informed consent was obtained from all patients.

## 3. Results

The following widely shared average values are reported for the dimensions of the human ocular bulb: (1) 23.5 mm transverse diameter, 23.2 mm vertical diameter, 24.2 mm antero-posterior diameter [12,13]. If we approximate the ocular bulb to an ideal sphere of 24 mm in diameter (d) and 12 mm in radius (r), the length of its circumference (C_E_) is equal to:C_E_ = 2πr = 2 × 3.14 × 12 = 75.36 mm (1)

The base of the ciliary body measures 6 mm in length, so the length of two ciliary bodies is 12 mm. The average white-to-white (WTW) distance is 12 mm. The corneal arch subtended to the WTW distance (C_C-WTW_) is approximately one-sixth of the entire corneal circumference, which would be:C_C-WTW_ ≈ 1/6 C_E_ ≈ 75.36/6 ≈ 12.56 mm. (2)

Adding up the corneal arc plus the length of the two ciliary bodies, we obtain:C_AS_ = C_C-WTW_ + 2C_CB_ ≈ 12.56 mm + 12 mm ≈ 24.56 mm, (3)
where C_AS_ is the linear measure of the arc of the spherical cap representing the anterior segment of the eye. The linear length of the theoretically photographable retina (CR) is then obtained by subtracting the linear measure of the anterior segment from the circumference of the ocular bulb:C_R_ = C_E_ − C_AS_ ≈ 75.36 − 24.56 ≈ 50.80 mm. (4)

In degrees, the angular extension of the theoretically photographable retinal surface (α_R_) can be obtained solving the proportion:C_E_:360 = C_R_:α_R_, i.e., 75.36:360 = 50.80:α_R_.(5)

Therefore:α_R_ = 242.68 degrees. (6)

The angular extension of the internal surface of the eye that cannot be photographed (α_AS_), corresponding to the anterior segment, can be calculated as:α_AS_ = 360 − α_R_ ≈ 360 − 242.68 ≈ 117.32 degrees (7)
and corresponds to the arc of spherical cap representing the anterior segment of the eye. The surface (S_E_) of an ideal spherical eye with a diameter of 24 mm is:S_E_ = 4πr^2^ = 4 × 3.14 × 122 = 1808 mm^2^. (8)

The measurement of the theoretically photographable retinal surface (S_R_) can be obtained with the proportion:S_E_:360 = SR:αR i.e., 1808:360 = S_R_:242.68. (9)

Therefore:S_R_ = 1218.8 mm^2^. (10)

The percentage of retinal surface covering the inner side of the eye (S_R%_) can then be calculated with the proportion:1808:100 = 1218:S_R%_. (11)

Therefore:S_R%_ = 67.4%. (12)

All results are summarized in Table 1.

We captured and analyzed fundus photographs by using a Zeiss Clarus 500 fundus camera. Performing a single shot, centered on the macula, it was possible to photograph a retinal linear length of 27.25 mm and a retinal surface area of ~570 mm^2^, which, according to the manufacturer [14], corresponds to an FOV of 133 × 133 degrees (very close to our geometrical model of 130.17 degrees) (Figure 2). Performing two shots with an automatic montage, we obtained a maximum retinal linear length of 42.34 mm, which corresponds to an FOV of 200 × 133 degrees wide by tall (202.42 × 130.17 degrees according to our model) (Figure 3). Performing four shots with an automatic montage, we obtained a maximum retinal linear length of 41.39 mm, an FOV of 200 × 200 degrees (197.73 × 197.73 degrees in our model), and a retinal surface area of ~1100 mm^2^ (Figure 4). Finally, performing six shots with a semi-automatic montage, we obtained a maximum retinal linear length of 49.46 mm and a maximum FOV of 236.27 degrees (according to our model). The retinal surface covered by six shots was 1406.39 mm^2^ (Figure 5).

## 4. Discussion

WF and UWF imaging allows the visualization of a significantly larger retinal area than previous techniques. In our geometrical analysis, we found that the angular extension of the theoretically photographable retina is 242 degrees, encompassing a surface of 1218.8 mm^2^ (67.4% of the inner surface of the eye) and a linear length of 50.80 mm.

According to the manufacturer [14], single-shot WF images captured using a Zeiss Clarus 500 have an FOV of 133 degrees, which includes about 570 mm^2^ (54.95%) of the retinal surface, allowing visualization up to the vorticose veins. For comparison, a traditional fundus camera has an FOV of 45 degrees, encompassing a retinal surface area of only 200 mm^2^. WF imaging is also a remarkable improvement compared to the 75 degrees obtainable with the 7FS, corresponding to about 375 mm^2^ (31.9%) of the retinal surface. In order to capture the same retinal area of an UWF device, we performed two shots and an automatic montage, obtaining an FOV of 200 × 133 degrees, which allowed the visualization of the far periphery, beyond the vorticose veins. The maximum FOV we could obtain was 236.27 degrees by performing six shots and a semi-automatic montage, which is almost the entire surface of the theoretically photographable retina (angular extension of 242.68 degrees). There is still a very small portion of the far peripheral retina that is impossible to capture, even with UWF imaging. The photographable retinal surface with six shots, as reported from the device, was 1406.39 mm^2^. This value is higher than the theoretically photographable retinal surface we calculated in our geometrical evaluation. Nagra et al. evaluated the human retinal surface area by ocular magnetic resonance imaging (MRI) of 73 eyes: the mean total retinal surface area was 1363 ± 160 mm^2^, which is also larger than our geometrical outcome [15].

A limitation of our analysis is that the comparison between the ideal eye and WF images was performed on different patients, even though all of them were healthy males, similarly aged, and had emmetropic eyes. We are currently recruiting patients to conduct a similar geometrical comparison in a wider cohort of patients, performing a more accurate statistical analysis. However, this methodological limitation is not sufficient to justify the aforementioned discrepancies. The most probable reason for the underestimation of the true anatomical measures of the retina is the approximation of the eyeball to an ideal sphere of 24 mm in diameter.

Furthermore, there are inherent issues with obtaining reliable linear retinal measurements when evaluating the peripheral retina. Drasdo et al. first studied the relationship between the input visual angles and linear distances measured along the surface of the retina [16]. By progressively increasing the distance of the input visual angle from the optic axis, a linear, directly proportional increase in the underlying retinal arc length is obtained. However, when considering very peripheral input visual angles that project beyond the equator, this function becomes non-linear, showing a modest droop in the retinal arc length. While having a negligible effect on the assessment of the posterior pole, this non-linear relationship may have a significant impact on the assessment of ocular diseases affecting the peripheral retina. These limitations may be even more impactful when considering eyes with a high degree of axial myopia or hyperopia.

Further insight was recently provided by Simpson et al., who identified major issues in angular retinal scaling and in the different methods of assessing the retinal FOV [9]. For instance, the FOV angle can be subtended at (1) the exit pupil, (2) the nodal point, or (3) the center of the retinal sphere, leading to significant discrepancies in its measurement.

Going forward, it would be desirable to choose a standardized unique method for the assessment of retinal FOV imaging, preventing harmful misunderstandings. A possible solution would be to follow the existing rules of the normative ISO 10940:2009 [11] for easier comparison with non-widefield imaging. We believe this would be highly beneficial for the future of WF and UWF imaging in clinical practice. Several authors previously described the impact of WF and UWF imaging on ophthalmologic diagnostics including uveitis [17], vasculitis [18], skip areas in the retinopathy of the premature [19], differential diagnosis between nevus and malignant melanoma [20], Coats disease, von Hipple Lindau syndrome, retinal detachment, and others. However, WF and UWF have arguably their most widespread application in the diagnosis and follow-up of diabetic retinopathy (DR). It is well known that eyes affected by DR may show peripherical retinal lesions associated with an increased risk of disease progression.

Verma et al. recently conducted a multicentric, perspective, observational study in India, obtaining UWF pseudocolor images of 715 patients (1406 eyes) affected by DR, using an Optos Daytona Plus (Optos plc, Dunfermline, Scotland, UK). The ETDRS grid was overlaid on stereographic projections of the UWF images. Both number and dimensions of retinal lesions were evaluated, showing two recognizable disease patterns: predominantly central lesions (PCL) and predominantly peripheral lesions (PPL). Eyes were included in the PPL group if >50% of the lesions were seen in at least one peripheral field as compared with the corresponding ETDRS field. The PPL pattern occurred in 37% of cases, while the PCL pattern occurred in 63%. The frequency of PPL varied significantly across all severity levels: 30.9% for mild non-proliferative DR (NPDR), 40.3% for moderate NPDR, 38.5% for severe NPDR, and 34.9% for proliferative diabetic retinopathy (PDR) [21].

Despite recent evidence regarding the prognostic value of peripheral lesions, current DR severity scales are still largely based on retinal microvascular lesions visible with 7FS photography, within an FOV of only 75 degrees. Logically, it would be advisable to take advantage of new imaging technologies, rethinking and updating the severity scores for DR based on retinal periphery. Significant efforts are being made to quantify DR lesions in UWF imaging and fluorescein angiograms, looking to generate objective classification metrics, with more reliable predictions regarding prognosis and the response to anti-VEGF treatment [16]. However, there are substantial obstacles to the diffusion of new scales based on wide-field imaging: principally, the high cost of WF/UWF devices and the lack of standardization for these imaging techniques, which may lead to partial discrepancies between different imaging systems [22].

## 5. Conclusions

The transition from 7SF to wide-field imaging is a true milestone in the diagnosis of chorioretinal pathologies. The angular extension of the photographable retina with these new devices is about three times larger. Considering the necessary approximations, our geometrical analysis showed that the angular extension of the theoretically photographable retina is about 242 degrees. Single-shot WF images, centered on the macula, captured using a Zeiss Clarus 500 have a FOV of 133 degrees (570 mm^2^), which includes about half of the retinal surface. When performing a semi-automatic montage of 6 different shots, we could obtain a maximum FOV of about 236 degrees (1400 mm^2^), which consists of almost the whole retina.

WF and UWF imaging allow detailed visualization of the retinal periphery, which is particularly impactful in the diagnosis and management of diabetic retinopathy. New severity scales should be adopted based on these imaging technologies.

WF and UWF imaging are revolutionary technologies in ophthalmology and should be promptly integrated as an indispensable part of the eye examination for their diagnostic, prognostic, and forensic value.

## Figures and Tables

**Figure 1 life-13-00202-f001:**
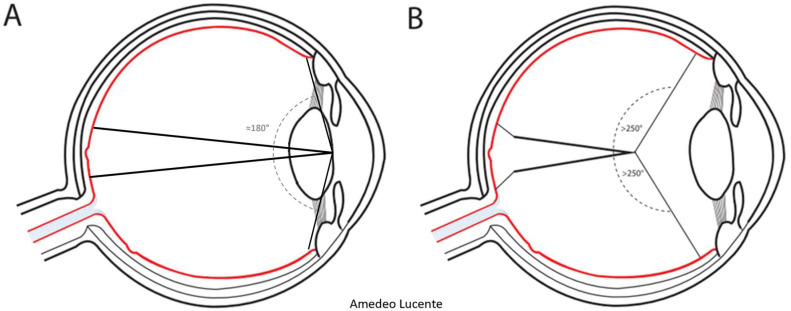
Methods for assessing the angular field of view (FOV) in retinal imaging. (**A**) The FOV assessed by visual-angle (θv). The FOV angle is subtended by the imaged retinal region at the exit pupil of the eye, according to ISO 10940:2009 standards. (**B**) The FOV assessed by eye-angle (θe). The FOV angle is subtended by the imaged retinal region at the spherical center of the eye, that is, the insertion of the vertical diameter of the eyeball and the visual axis.

**Figure 2 life-13-00202-f002:**
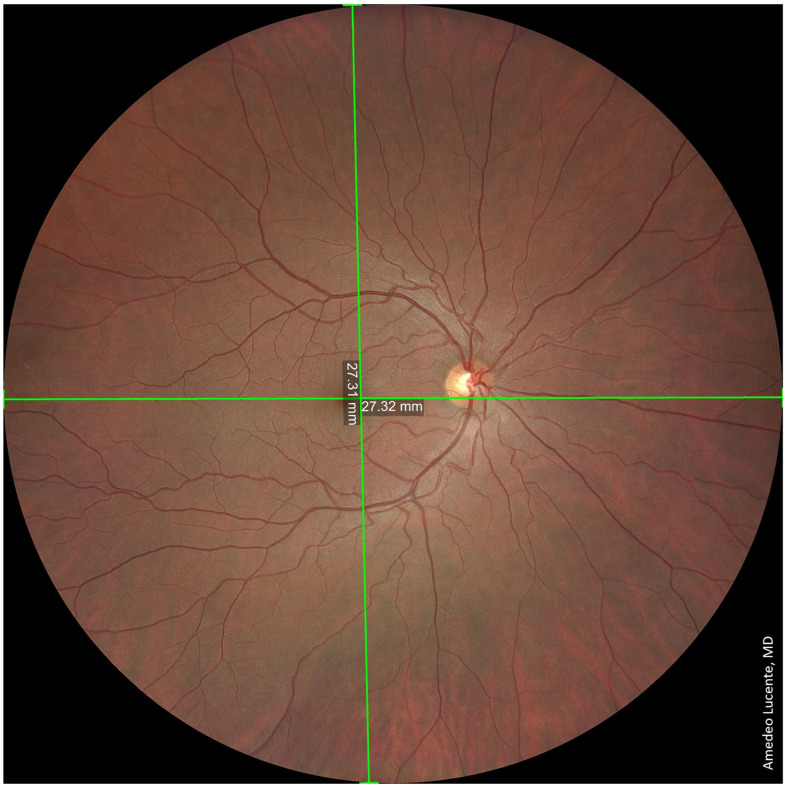
A single shot, centered on the macula. Photographable retinal linear length = 27.25 mm. FOV ≈ 133 × 133 degrees. Retinal surface ≈ 570 mm^2^.

**Figure 3 life-13-00202-f003:**
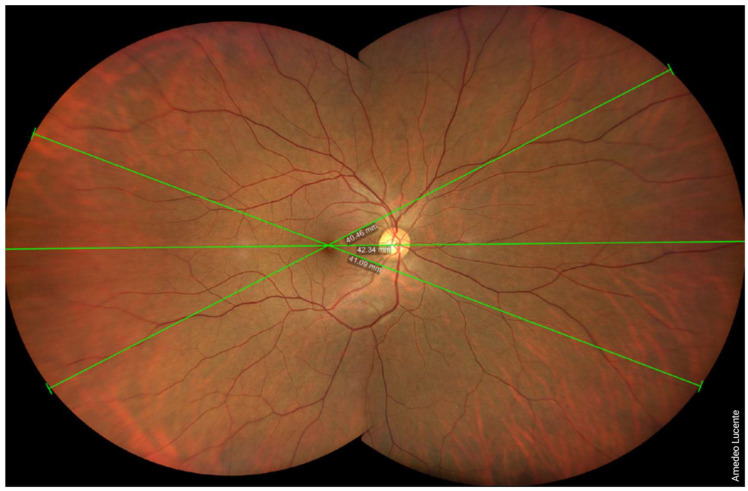
Two shots, combined by automatic montage. Maximum photographable retinal linear length = 42.34 mm. FOV ≈ 200 × 133 degrees (wide by tall).

**Figure 4 life-13-00202-f004:**
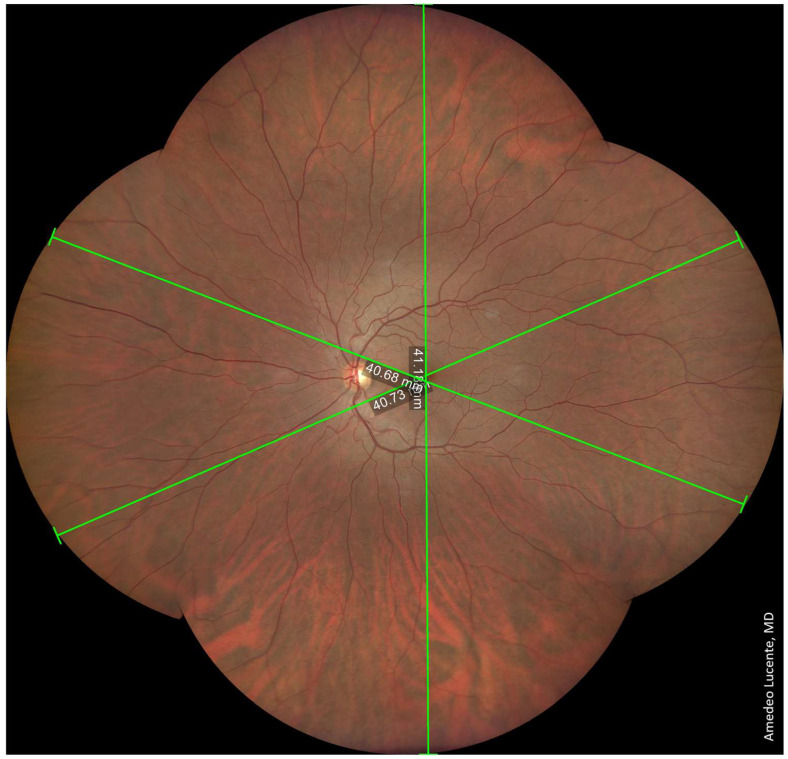
Four shots, combined by automatic montage. Maximum photographable retinal linear length = 41.39 mm. FOV ≈ 200 × 200 degrees (wide by tall). Retinal surface ≈ 1100 mm^2^.

**Figure 5 life-13-00202-f005:**
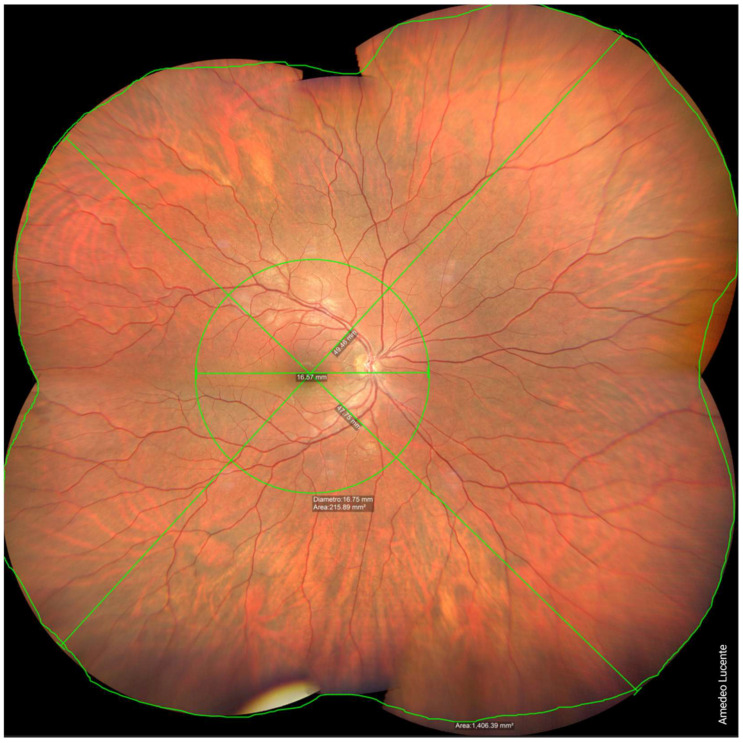
Six shots, combined by semi-automatic montage. Maximum photographable retinal linear length = 49.46 mm. Maximum FOV ≈ 236.27 degrees. Retinal surface ≈ 1406.39 mm^2^.

**Table 1 life-13-00202-t001:** Measures of an ocular bulb approximated to an ideal sphere with a 24 in diameter.

Circumference of the eye	C_E_ = 2πr = 75.36 mm
Length of the base of the ciliary body (C_CB_)	C_CB_ ≈ 6 mm
White-to-white (WTW) distance	WTW ≈ 12 mm
Corneal arc subtended to the WTW distance (C_C-WTW_)	C_C-WTW_ = 1/6 C_E_ ≈ 12.56 mm
Length of the arc of spherical cap representing the anterior segment (C_AS_)	C_AS_ = C_C-WTW_ + 2C_CB_ ≈ 24.56 mm
Linear length of the theoretically photographable retina (C_R_)	C_R_ = C_E_ − C_AS_ ≈ 50.80 mm
Angular extension of the theoretically photographable retina (α_R_)	C_E_:360 = C_R_:α_R_ α_R_ = (C_R_ × 360)/C_E_ ≈ 242.68 degrees
Angular extension of the anterior segment (α_AS_)	α_AS_ = 360 − α_R_ ≈ 117.32 degrees
Surface of the eye (S_E_)	S_E_ = 4πr^2^ = 1808 mm^2^
Area of the theoretically photographable retinal surface (S_R_)	S_E_:360 = S_R_:α_R_ S_R_ = (S_E_ × α_R_)/360 ≈ 1218.8 mm^2^

## Data Availability

Not applicable.

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
