# Peer review of "Widefield and Ultra-Widefield Retinal Imaging: A Geometrical Analysis"

_life, 2023, doi:10.3390/life13010202_

Round 1
Reviewer 1 Report
Widefield imaging is spreading and I think more knowledge should be published on the topic. The new measurements will help unifying the data among papers
I think it should be published
Author Response
WF retinal imaging and FOV measurement still remain controversial topics in literature, with ambiguous terminology and a lack of standardized methods of assessment. We added more information about this subject in both Introduction and Discussion, highlighting the importance of choosing a unique method for the evaluation of WF retinal imaging, preventing harmful misunderstandings that may have negative repercussions on clinical practice.
Reviewer 2 Report
The main point of this paper is very clear, that this type of measurement is available, and that it is very valuable in clinical practice.
The paper also discusses the background to the images, and how they are scaled, but there is a lot of confusing information in the literature, and it is not easy make things clear. The discussion in this paper is really about the plausibility of the retinal scaling, rather than directly verifying what the camera is reporting. And that seems very reasonable, but perhaps it should be stated clearly, along with any important assumptions.
And there just seems to be 1 patient, with no patient details. Is this a fairly average patient, who might be somewhat close to an average eye? Perhaps some details could be added.
It is hard to find a clear description of the scaling for wide-angle images of the eye, but there are various publications that give hints. The Clarus 500 has “Field of view (measured from the center of the eye)” on the Zeiss website. With no real explanation for what that means.
One paper that specifically discusses scaling is: XINCHENG YAO, DEVRIM TOSLAK, TAEYOON SON, AND JIECHAO MA. Understanding the relationship between visual-angle and eye-angle for reliable determination of the field-of-view in ultra-wide field fundus photography. Vol. 12, No. 10 / 1 Oct 2021 / Biomedical Optics Express 6651-6659
This one by similar authors uses an indirect ophthalmoscopy lens to generate an initial image: DEVRIM TOSLAK, FELIX CHAU, MUHAMMET KAZIM EROL, CHANGGENG LIU, R. V. PAUL CHAN, TAEYOON SON, AND XINCHENG YAO. Trans-pars-planar illumination enables a 200° ultra-wide field pediatric fundus camera for easy examination of the retina. Vol. 11, No. 1 / 1 January 2020 / Biomedical Optics Express 68-76
This is less relevant because of its complex imaging system, but it is related: Shuibin Ni, Thanh-Tin P. Nguyen, Ringo Ng, Shanjida Khan, Susan Ostmo, Yali Jia, Michael F. Chiang, David Huang, J. Peter Campbell, AND Yifan Jian. field of view non-contact handheld swept-source optical coherence tomography Vol. 46, No. 23 / 1 December 2021 / Optics Letters Letter 105
If you try to evaluate the background, this older paper that is available online (Researchgate?) has some background. Input angles to the eye are related to linear distances along the retinal surface: Drasdo, N.; Fowler, C.W. Non-linear projection of the retinal image in a wide-angle schematic eye. Br. J. Ophthalmol. 1974, 58, 709–714.
A recent evaluation relates 3 different methods of scaling retinal angles, where the angle start point might be at (a) the exit pupil, (b) the nodal point, or (c) the center of the retinal sphere. These all show up in the literature in various places. There is a strong linear relationship with angle to very large angles, but it seems to fall off at the very wide-angle angles, where it is not linear any longer. Simpson MJ. Scaling the Retinal Image of the Wide-Angle Eye Using the Nodal Point. Photonics. 2021; 8(7):284. https://doi.org/10.3390/photonics8070284
Another wrinkle was added to the widefield limit in this paper, which added a discussion of Gott’s flat map, which was discussed for mapping the earth a couple of years ago: Simpson MJ, "Nodal points and the eye," Appl. Opt. 61, 2797-2804 (2022)
Another paper that has some relevance is one where a model eye is used to evaluate a retinal camera system. Using a flat surface in air to test a camera seems very limited in comparison, if the whole purpose is to image a highly curved surface. The model here only extends to a hemisphere (or 180 degs for angles from the center). There don’t seem to be any significant citations of it either. Anthony Corcoran, Gonzalo Muyo, Jano van Hemert, Alistair Gorman & Andrew R. Harvey (2015) Application of a wide-field phantom eye for optical coherence tomography and reflectance imaging, Journal of Modern Optics, 62:21, 1828-1838, DOI: 0.1080/09500340.2015.1045309
This is all too much to fully sort out in one paper, but perhaps the authors could add in additional references to the introduction or discussion, and make sure that the terms are crystal clear. In the Yao et al reference they use two terms: Visual angle FOV, and Eye angle FOV. If there are better ones, they could be used instead, but it is a huge issue if a term FOV actually has 2 different meanings. Anything the authors can do to make this clearer would be beneficial.
Fundus imaging, and indirect ophthalmoscopy, have been around for so long that there does not seem to be a clear reference for what is going on. The Drasdo et al reference captures an underlying concept, but somewhat indirectly, without emphasizing how phenomenal it actually is. Input angles to the eye end up as linear distances along the surface of the retina. Mostly linear, and when you get beyond the equator there is a droop. And at very large angles there may be more of a droop, but also the Drasdo paper has a very simple eye, and it may not be modeling correctly. Though the other papers give a similar general result. Vision people deal with visual angles, so this can relate input angles to retinal locations. Retinal people look more for retinal features, with a more flexible relationship to input angle, perhaps. If everything was exactly linear though, why wouldn’t visual angles always be used.
And the goal of the camera, or an ophthalmoscopy lens, is probably to take angles leaving the eye, and then convert them linearly with angles to radial locations on a flat surface. But is that what the camera is trying to do, and how well does it actually do that? It is also not just the angle that is important, the image also has to be in focus. And what happens when the angles go past the retinal hemisphere (which Gott’s flat map raises questions about, because a flat map of the earth’s surface is somewhat similar to a flat image of the retina).
The calculations here evaluate the plausibility of the images that were obtained, and that seems fine. As long as the broader background is addressed. This is also for an average eye, and there would be variations for other eyes? And clinically it is perhaps fine to not have perfect linearity when evaluating the retina as long as the image details are clear? If so, perhaps that could be said also.
A couple of other specific comments:
Fig. 1 The angles are not clear. Perhaps the 180 degs relative to the center is being mentioned in 1A?, but there are 2 180 values for the other location.
Foramen is not a word normally used in English for this. It is the center of the exit pupil? Or the real pupil?
The distances describing the eye that are discussed in the text could be depicted on a drawing in Fig 1.
